# Mental Health of Chinese Online Networkers under COVID-19: A Sociological Analysis of Survey Data

**DOI:** 10.3390/ijerph17238843

**Published:** 2020-11-28

**Authors:** Yang Xiao, Yanjie Bian, Lei Zhang

**Affiliations:** 1School of Philosophy and Government, Shaanxi Normal University, Xi’an 710119, China; xiaoyang2017@snnu.edu.cn; 2Institute for Empirical Social Science Research, Xi’an Jiaotong University, Xi’an 710049, China; 3Department of Sociology, University of Minnesota, Minneapolis, MN 55455, USA; 4Department of Sociology, University of Colorado Colorado Springs, Colorado Springs, CO 80918, USA; lzhang4@uccs.edu

**Keywords:** mental health, socioeconomic status, lifestyle, social capital, COVID-19

## Abstract

This paper reports the results of a recent survey of Chinese WeChat networkers (*n* = 2015, August 2020) about China’s mental health conditions under COVID-19. The purpose of the survey was to measure symptoms of depression, anxiety, and somatization by using a standard 18-item battery and assess how the results were related to an individual’s socioeconomic status, lifestyle, and social capital under an ongoing pandemic. The survey reveals that the pandemic has had a significant impact, as the respondents had more serious mental symptoms when their residential communities exhibited a greater exposure to the spread of the virus. The socioeconomic status of the respondents was negatively associated with the mental symptoms. It modified the impact of COVID-19, and its effect was substantially mediated by measures of lifestyle and social capital.

## 1. Introduction

The outbreak of the COVID-19 pandemic since December 2019 has created a serious and ever-challenging public health crisis around the world. As of 10 November 2020, globally, there have been 52 million total confirmed cases and 1.28 million deaths. In light of the unavailability of either a vaccine to resist the virus or special therapy to treat the affected, mental suffering from COVID-19, among other consequences, has elevated to a level so high that it has generated widespread concerns from members of the general public, medical professionals, and public officials, including the United Nations General Secretary [1]. This paper presents the findings of a recent survey (August 2020) designed to assess the level and pattern of Chinese people’s mental health under the COVID-19 pandemic.

Our analysis of the survey data is inspired by both our observation of China and sociological literature that calls for attention to be paid to social variables of health inequality. The earliest COVID-19 cases were reported in Wuhan, which is a large metropolis in central China, in late December 2019. The virus quickly spread to other cities and regions of the country, but around early April, the spread of the virus was brought under control in China after more than two months of continuous efforts to forcefully apply the measures of city and community lockdowns, mobility restrictions, facial mask wearing, and social distancing [2]. A large-scale survey in April showed that Chinese people varied in their attitudinal and behavioral responses to COVID-19, resulting in measured inequalities in physical health and subjective well-being by education, occupation, income, and social relations [3]. However, this study did not report on mental health. Mental health research from around the world, on the other hand, has revealed that an individual’s socioeconomic status is correlated with a stream of symptoms [4,5], such as depression, anxiety, and somatization [6,7]. More recent studies have shown that one’s socioeconomic status is correlated with lifestyle and social capital, and together, these variables jointly increase subjective well-being [8,9]. To conduct a sociological analysis of Chinese people’s mental health under COVID-19, we were thus inspired by prior research to apply three concepts to help uncover and explain observed variations in mental health: Socioeconomic status; lifestyle; and social capital.

## 2. Theoretical Considerations

### 2.1. Socioeconomic Status and Mental Health

A hallmark in the study of public health is Britain’s Black Report, which has sparked a lively and constructive debate over the past four decades [4]. While the Black Report revealed a positive correlation between one’s socioeconomic status (SES) and mental health, which has since been widely confirmed in studies across the globe [5,10,11,12,13,14,15], researchers have debated about the underlying causal direction of that correlational relationship. Some researchers maintain that health is related to socially determined structural factors, such as SES [16], whilst others contend that social mobility, which is an important mechanism affecting people’s positions in the social structure, is affected by health in ways in which the healthy move up the class hierarchy, while the less healthy move down it [17]. Additionally, others have established a two-way causal model [18,19,20] and found common causes for SES and health [21,22,23].

Researchers have offered plausible explanations for why SES is related to health. First, knowledge obtained from a good education leads to a healthy lifestyle, helping to prevent unhealthy behavior and supporting people in rationally coping with mental problems [24,25]. Second, a higher income increases people’s financial ability to afford a spending budget for the living standards and professional services that are required to maintain healthy lifestyles [26], even if too much money tends to create health problems that lower happiness [27]. Finally, one’s occupational status and mental health are correlated with one another. For example, manual and nonmanual jobs vary in working conditions, as measured by work autonomy, the laboring intensity, psychological pressure, and physical safety, which all affect occupants’ physical and mental health [28]. When education, income, and occupation are integrated into a composite SES scale, these results are also confirmed [19,29,30,31].

### 2.2. Lifestyle and Mental Health

Lifestyle is an important mechanism whereby SES is related to health. Social epidemiologists have studied how different lifestyles are associated with people’s exposure to health hazards, such as smoking, alcoholism, and an inadequate diet, all of which tend to deteriorate health conditions [8]. One critique of this research tradition is that it puts too much emphasis on health-damaging behaviors [32], calling for attention to be paid to physical exercises, routine physical examinations, and safe driving as health-promoting measures of lifestyle. Research has shown that SES is positively correlated with lifestyle [33], and high- and low-brow lifestyles, as measured by different tastes and preferences for consumer goods and cultural products [34], which result in a sharp difference in people’s health status [35].

Lifestyles do not merely mediate SES’s effect, but are also directly associated with one’s health status. For instance, Bourdieu’s research on habitus and class distinction in French society reveals a large within-group variation in lifestyles among people with similar levels of SES [34]. Empirical findings from Denmark [36], Australia [37], and the United States [38] have confirmed that with respect to SES’s effect, the greater people deviate from healthy lifestyles, the greater the risks and the lower the status of their health conditions. In Chinese society, unhealthy lifestyles are concentrated in populations with the lowest and highest SESs, as observed for the Chinese Mainland [39] and Hong Kong [40]. These results suggest that we must treat lifestyle as both an independent and mediating variable when analyzing the relationship between SES and mental health.

### 2.3. Social Capital and Mental Health

Social capital is another important mechanism through which SES is related to one’s health status. Social capital refers to the collective and interpersonal resources that are embedded in and can be mobilized from the networks of ongoing social relations [41,42,43], and standard measures of social capital include the name generator, position generator, and resource generator [9,44]. People with a higher SES tend to have greater social capital [45,46]. Social capital, however measured, is the primary source of social support for mitigating depression, anxiety, and other mental symptoms [47,48]. In the United States, for example, adolescents from urban neighborhoods with greater bonding social capital [49] tend to have lower levels of depression [50]; socially isolated individuals are less healthy, both psychologically and physically, than those that are well-connected [51]; and measures of formal and informal networks significantly increase mental health [52,53].

In China, research has revealed the importance of social capital for mental health. For example, core personal networks of kin relations and close friends are important for maintaining people’s mental health at a high level [54], especially when people with a lower SES have neither access to quality public services nor interest in public participation [55]. Under this circumstance, “who you know” has become a primary source of social support for Chinese people [56], and those in authority positions to whom people are connected matter greatly for their mental health and well-being [57]. More recently, an analysis of the 2017 module data of the International Social Survey Programme showed that, compared to Europeans and North Americans, Chinese citizens are happier when they are bonded to society at large through informal, rather than formal, networks [58].

### 2.4. Propositions

The insights learned from the literature review lead us to propose the following propositions that will guide our data analysis.

**Proposition** **1.**
*COVID-19 has a significant impact on mental health. The spread of the virus is uneven and therefore, the severity of its effect varies across localities and communities. Some communities are heavily hit and have both confirmed cases and deaths; other communities are less heavily hit and have fewer confirmed cases, but no deaths; and still other communities are not hit as badly and have no confirmed cases at all. Therefore, there is every reason to believe that, all other factors being equal, people’s mental health conditions decrease as they and their surrounding social environments (family, neighborhoods, and networked communities) are affected more severely by the virus.*


**Proposition** **2.**
*SES is related to mental health. Specifically, persons with a higher SES tend to be in a more informed (education), more capable (income), and less risky (occupation) position to cope with COVID-19, and are thus better able to prevent or minimize its negative impact and maintain a higher mental health status than their counterparts with a lower SES.*


**Proposition** **3.**
*This proposition is about the mediating effects of lifestyle and social capital on mental health. First, persons pursuing healthy lifestyles are more likely to have developed healthy habits, and are thus better able to prevent or minimize the negative impact of COVID-19 and maintain higher mental conditions than their counterparts with unhealthier lifestyles. Second, persons with greater social capital are more likely to obtain social support from connected individuals, and are thus better able to prevent or minimize the negative impact of COVID-19 and maintain a higher mental status than their counterparts with lower social capital.*


**Proposition** **4.**
*SES modifies COVID-19’s impact on mental health. If one is highly educated or wealthy, it is easier to maintain a paid job or handle sudden job loss from COVID-19, which might otherwise lead to severe stress and depression. This modifying effect is expected to be stronger under a more severe COVID-19 situation, when there is a greater urgency to maximize SES-embedded capacities and resources to combat the virus than in a less severe COVID-19 situation.*


## 3. Materials and Methods

### 3.1. Data

We conducted a survey of WeChat networkers from 27 to 30 August 2020. At this point, the outbreak of COVID-19 in China, although still ongoing, had been brought under a mode of “normative control and prevention” since late April. This was the context in which our survey was conducted. When the COVID-19 threat prevented data collection from face-to-face interviews, China’s WeChat network became a second-best sampling frame for an online survey. Developed by Tencent Holdings Limited in Shenzhen, China in 2011, by the first quarter of 2020, the WeChat network had 1.2 billion monthly active users [59], or 86% of China’s total population (1.4 billion). Of the most active WeChat networkers, 57.7% are females, 78% are aged 18-30, and 87% are college educated or higher individuals [60], representing the younger and highly educated segment of China’s adult population.

With the professional assistance of a WeChat survey company based in Shenzhen, neighboring Hong Kong, we designed a short module with an average survey time of ten minutes. Participation in this survey was voluntary, and participants could opt out at any time. Moreover, participants were assured anonymity and confidentiality of their responses. The survey was granted ethical approval by Shaanxi Normal University (202002001). Each respondent was offered 8 Chinese yuan (equivalent to $1.2) for a successful survey. To ensure data quality, only one completed questionnaire could be submitted per IP address, and two human-authentication questions were used in the middle of questionnaire to screen out computer-generated, program-coded answers intended to collect the payment. We controlled for gender balance, but did not control for any other personal attributes of respondents. Out of a total of 3491 participants, 2015 completed questionnaires were valid, resulting in a response rate of 57.72%. The valid respondents were from all provinces except Tibet. Due to an extremely small population size, Tibetans are rarely represented in national random sample surveys, including the well-known Chinese General Social Survey [61].

### 3.2. Measures

Mental health was the dependent variable in our study. It was measured by using the standard Brief Symptom Inventory-18 (BSI-18) [7], which is an 18-item battery employed for measuring psychological symptoms. Space and time considerations regarding our online survey meant that we had to reject an alternative plan of using a longer measuring approach, such as a 90-item battery [6] or a 53-item battery [62]. Following Derogatis [7], we constructed three Likert-scale variables, each of which had six contributing items: Depression; anxiety; and somatization. Since all items were measured with a five-point scale, ranging from 1 (not at all) to 5 (very much), each of our reconstructed variables had a value-range of 5 to 30.

The severity of COVID-19 was a contextual variable in our analysis. This refers to the extent to which a residential community (a rural village or urban neighborhood) exhibited differential exposure to COVID-19. We did not have official data on the numbers of confirmed cases and deaths across localities and communities around the country. Instead, our survey respondents were asked whether their residential communities had confirmed COVID-19 cases and deaths. Since this information was publicly circulated within residential communities, our respondents had no problem answering the question. Based on their responses, the respondents’ residential communities were classified as one of three levels of COVID-19 severity: “Low severity” refers to a community in which no confirmed COVID-19 cases were identified; “medium severity” refers to a community with confirmed COVID-19 cases, but no deaths; and “high severity” refers to a community with both confirmed COVID-19 cases and at least one death.

The socioeconomic status (SES) was the main independent variable in our study. We followed a long tradition of social stratification and mobility research [63,64] by merging a respondent’s education, occupation, and family income into a composite scale of SES through a factor analysis. Education was measured by years of schooling. Occupation was divided into five rank-order categories, from low (1) to high (5): Jobless; students; blue-collar workers; white-collar workers; and employers. Because a fifth of WeChat networkers do not earn an income, we measured respondents’ family annual income in 2019. While the results of factor analysis are presented in the next section, here, we note that the higher the factor score, the higher the respondent’s SES.

Lifestyle was both an independent and mediating variable in our regression analysis. Lifestyle is a multidimensional concept, which has been measured in prior research by consumerist perceptions [65], preferable time allocation [66], values toward collectivism vs. individualism [67,68], and behavioral orientations towards healthy vs. unhealthy habitus [69]. Our survey included six questions on aspects of lifestyle that are most relevant to the Chinese context, and a factor analysis generated a three-dimensional solution, with each dimension being measured by two items: (1) Health-damaging behavior, measured by two items about the frequency of smoking and frequency of drinking; (2) health-promoting behavior, measured by the frequency of physical exercise and frequency of regular physical checkups; and (3) a valuation of individualism, measured by the sense of privacy and freedom of expression.

Social capital was also both an independent and mediating variable in our regression analysis. As reviewed above, social capital can be measured in multiple ways [9,44]. Encouraged by relevant research on China [3,70] and due to the space and time constraints of our online survey, we narrowed the concept’s scope down to two network dimensions: (1) Network intensity refers to the extent of intimate interactions one has with others within and around the family during COVID-19, including spousal relations and intergenerational relations within the family, and (2) network extensity refers to the extent of interactions one has with more distant individuals, well beyond the family and intimate circles, including online interactions with significant others, the frequency of online activity, and the number of friends one socializes with online on a daily basis. As measured, these two dimensions can be conceptualized as bonding social capital and bridging social capital, respectively [49].

Control variables were measured by a set of social and demographic attributes of the respondents. These include gender (a dichotomy, coded in regression analysis for male = 1 and female = 0), age (a continuous variable), marital status (a dichotomy, married = 1 and otherwise = 0), religious belief (a dichotomy, yes = 1 and no = 0), residence (a dichotomy, urban = 1 and rural = 0), and membership of the Chinese Communist Party (CCP, a dichotomy, yes = 1 and no = 0). In Western countries, political party affiliation is a personal choice and related to people’s social-political values. In China, research has shown that CCP members are a selective group of elites and quasi-elites and therefore, CCP membership is a marker of political screening and power, as well as socioeconomic achievements [3,30,70]. While CCP is highly related to SES in China, we included this variable as a statistical control to estimate the net effect of SES on mental health.

### 3.3. Methods of Statistical Analyses

Statistical analyses were performed using STATA (Version 14, StataCorp, College Station, TX, USA) or SPSS (Version 27, IBM Corp, Armonk, NY, USA). We took four steps to conduct our statistical analyses. First, we obtained descriptive information of all variables (Table 1) and conducted factor analysis to generate the factors for SES, lifestyle, and social capital (Table 2). Second, we conducted bivariate analyses for testing Proposition 1 about the variation of mental symptoms across levels of COVID-19 severity. Since mental symptoms were interval variables and COVID-19 severity was a categorial variable, we used the ANOVA model, Scheffe’s test, and Chen’s method of standardization to detect the mean differences in depression, anxiety, and somatization across three levels of COVID-19 severity (Figure 1 and Table 3). Third, for testing propositions 2 and 3, we used Ordinary Least Square (OLS) multiple regressions to estimate the effects of SES, lifestyle, and social capital on each of the continuous measures of mental symptoms with statistical controls (Table 4, Table 5, Table 6 and Table 7). Finally, we used OLS regressions with interaction terms to test Proposition 4 about the extent to which SES modifies the impact of COVID-19 severity on mental symptoms.

## 4. Results

### 4.1. Descriptive Findings

The left panel of Table 1 presents the percentage distributions of our 2015 survey respondents according to social-demographic variables. The gender composition is balanced. The age distribution is skewed toward younger age groups of 18–29 (62.68%) and 30–39 (30.72%), which is highly representative of the population of the most active WeChat networkers we previously described in the methodology section. More than 71% of the respondents were married; 14% had a religious belief; there were more urban than rural residents (55% vs. 45%); and about 18% were members of the Chinese Communist Party, which is China’s leading political party. These were applied as control variables in our regression analyses.

As shown in the right panel, the educational distribution is skewed toward college educated (70.07%) and above (6.65%) individuals, with a minority gaining only the compulsory education at the levels of senior high school (20.45%) and junior high school or below (2.82%). Close to 9% were small employers (hiring 50 or fewer employees, but no larger business owners were found), more than 42% were white-collar employees, close to 25% were blue-collar workers, close to 21% were college students still in school, and about 4% were jobless people. The average family income was 176,733 Chinese yuan (or $26,000), with a large standard deviation of 232,867 yuan, and the resulting coefficient of variation of 1.32 (SD/mean) indicates a high degree of income inequality in China.

By the time of our survey in late August 2020, China’s numbers of daily confirmed COVID-19 cases were in the single and double digits, with the total number of infections being 85,000 and total deaths being 4600. These numbers imply an extremely low exposure of Chinese people (1.4 billion) to the virus, or six confirmed cases for 100,000 people. However, our survey respondents indicated that their residential communities displayed differential exposure to COVID-19: close to 9% of the respondents lived in high-severity communities, where both confirmed cases and deaths were identified; about one fourth in medium-severity communities, where some confirmed cases were known to the respondents; and two thirds in low-severity communities, where no confirmed cases were reported.

The means and standard deviations of the three mental symptoms were as follows: Depression, 9.14 ± 4.08; anxiety, 8.77 ± 3.94; and somatization, 7.65 ± 2.73. Song et al. [71] conducted a cross-sectional online survey of China’s working adults (*n* = 709) on mental health from 9 to 22 April 2020. They used different scoring systems to calculate Likert scales of depression (10 items, on a 4-point scale, with 0–3 value assignment for each item), anxiety (7 items, 0–3), somatization (12 items, 0–4), and insomnia (7 items, 0–4). They then used a self-determined cutting point to break each scale into a dummy variable to measure the “prevalence” of the measured psychological symptoms. Through converting our coding systems into theirs by applying a measurement standardization approach [72], we reconstructed our three Likert scales for the working adults of our sample (75.5%) to permit for a within-China comparison. In April, China’s prevalence of depression, anxiety, and somatization was 13.5%, 12.7%, and 20.7%, respectively. In August, it decreased to 6.5%, 5.0%, and 1.7%, correspondingly. This decreasing trend is suggestive at best, because the August sample’s working people were an average of 5 years younger than April’s (30 vs. 35).

### 4.2. Findings from Factor Analysis

We conducted three sets of principal-component factor analysis for our sociological variables, including SES, lifestyle, and social capital, and Table 2 presents the results of factory analysis. In all of these analyses, factors with eigenvalues greater than 1 were retained. The method of orthogonal varimax rotation was used to generate uncorrelated factor scores and other statistical results.

For SES, the factor analysis generated a one-factor solution, with 53% of variance explained (VE) and a KMO (Kaiser–Meyer–Olkin) value of 0.62, which is well above the threshold of 0.50. The factor loadings were 0.70, 0.73, and 0.75 for education, occupation, and income, respectively. This indicates a similar contribution of each of the three variables to the generated factor score, resulting in a standardized score with a maximum of 2.58 and minimum of −3.61.

For social capital, a two-factor solution was obtained, with 61% of variance being explained and a KMO value of 0.55. The first factor was the network intensity, and the two measuring items—spousal relations and intergenerational relations within the family—displayed similar contributions to the generated factor, as shown by the factor loadings (0.89 and 0.88, respectively). This factor score exhibited a narrow range (max = 2.06, min = −3.62), implying a centralized distribution of bonding social capital around the family. In contrast, the second factor of network extensity had a larger range (max = 2.15, min = −9.29). This factor was generated from three items about one’s distant connections with individuals online, which are sociologically characterized as bridging social capital. The results here seem to indicate that Chinese people vary more in bridging social capital than in bonding social capital.

For lifestyle, three factors were generated, with 68% of the variance being explained and a KMO value of 0.53. The right-hand panel of Table 2 shows that each factor has two measuring items, and factor loadings indicate that their contributions to the generated factor are similar. The factor for health-damaging behavior has a comparably higher maximum value (max = 2.75, min = −1.25) than health-promoting behavior (max = 1.72, min = −2.43) and valuation of individualism (max = 1.83, min = −2.77).

### 4.3. Mental Symptoms by COVID-19 Severity

As shown in Figure 1, the ANOVA model reveals significant group mean differences of depression across the three levels of COVID-19 severity (F-test score of 17.20, *p* < 0.001). Scheffe’s pairwise mean comparisons reveal that (1) mean depression of the high-severity community is 1.635 (*p* < 0.001) higher than that of the low-severity community; (2) mean depression of the medium-severity community is 0.818 (*p* < 0.001) higher than that of the low-severity community; and (3) there is no significant mean difference between high- and medium-severity communities. As for anxiety and somatization, the ANOVA models and Scheffe’s tests yielded consistently significant results in every pairwise comparison. These results support Proposition 1.

To facilitate future meta-analysis, we calculated the standardized effect size (known as Cohen’s d) by using “low severity” as the control group and treating “medium severity” and “high severity” as two levels of treatment. Technical details aside (Appendix A), Table 3 presents the standardized effect sizes (the last three columns for Cohen’s d scores). As can be seen, the higher the degree of COVID-19 severity, the stronger the symptoms of depression, anxiety, and somatization.

### 4.4. Regression Results on Depression

Table 4 presents the results from multiple OLS regressions of depression. Model 1 shows that age and marital status generate significant, negative coefficients. The age coefficient indicates that older persons tend to have weaker depression symptoms. Because 93% of our respondents were aged 18–39, the age effect actually means that the younger, socially less experienced tend to have stronger depression symptoms than the more mature middle-aged people under the pandemic. A separate analysis using the natural spline approach identified no higher-degree age effect (Appendix A). As for the negative effect of marital status, it means that married couples have significantly lower depression symptoms than unmarried individuals, which is a sign of marriage as a source of social and psychological support during the pandemic. Our interpretation from now on will be focused on the coefficients of the independent and intervening variables.

Model 1 was designed to estimate the statistical significance and effect size of COVID-19 severity dummies and SES. Compared to residents in communities with a low COVID-19 severity (mean depression of 10.41, as marked by the constant), residents in communities with a medium COVID-19 severity have higher depression symptoms by a margin of 0.97 (*p* < 0.001), and those in communities with a high COVID-19 severity have the strongest depression symptoms by a margin of 1.81 (*p* < 0.001). These are considerable and net effects of COVID-19 severity on depression, independent of individual attributes (control variables), confirming the findings from Figure 1 and Table 3 in support of Proposition 1.

As expected, SES exhibited a significant, negative coefficient. This shows that for a one-unit (standard deviation) increase in SES, depression will decrease by a margin of −0.52 (*p* < 0.001). The effect size of SES is reduced to −0.30 in Model 2, where the measures of lifestyle and social capital are added, but it retains its statistical significance. These findings support Proposition 2.

As can be seen in Model 2, SES’s effect on depression is mediated by lifestyle and social capital. First, each of the three variables of lifestyle generates a significant coefficient: A standard deviation increase in health-damaging behavior increases one’s depression by 0.31 (*p* = 0.001); a standard deviation increase in health-promoting behavior decreases one’s depression by −0.40 (*p* < 0.001); and a standard deviation increase in valuation of individualism decreases one’s depression by −0.22 (*p* = 0.007). Second, each of the two social capital variables also produces a significant coefficient: A standard deviation increase in network intensity decreases one’s depression by −1.24 (*p* < 0.001), and a standard deviation increase in network extensity decreases one’s depression by −0.34 (*p* < 0.001). The explained variance substantially increased from Model 1 (adjusted *R*^2^ of 0.063) to Model 2 (adjusted *R*^2^ of 0.167). These results provide evidence in support of Proposition 3.

### 4.5. Regression Results on Anxiety

Looking at Table 5, one can observe that the presented regression results reveal an overall similar pattern for anxiety as those for depression just discussed. Briefly, as shown in Model 3, age is negatively associated with anxiety, and people with religious beliefs tend to have a higher degree of anxiety than their counterparts without religious beliefs. Compared to residents in communities with a low COVID-19 severity, residents in communities with medium and high degrees of COVID-19 severity, respectively, tend to have a significantly higher anxiety score (1.10 and 2.18, respectively). These effects continue to be significant and retain their effect sizes in Model 4, in which other predictors are incorporated. Model 3 shows that SES has a significant direct impact on anxiety; a one-unit increase in SES will decrease anxiety by −0.28. This effect is reduced by half to an insignificant −0.14 in Model 4, in which the measures of lifestyle and social capital function as mediating variables. In this model, health-damaging behavior increases one’s anxiety by a margin of 0.29, health-promoting behavior decreases one’s anxiety by −0.15, valuation of individualism decreases one’s anxiety by −0.28, network intensity decreases one’s anxiety by −0.97, and network extensity decreases one’s anxiety by a margin of −0.27. The adjusted R-squared value substantially increases from 0.051 in Model 3 to 0.115 in Model 4. All these effects are statistically significant after statistical controls, providing another set of evidence in support of propositions 1–3.

### 4.6. Regression Results on Somatization

As shown in Table 6, somatization, which is a human phenomenon that attributes one’s mental health problems to physical health symptoms, reveals a similar pattern to those for depression and anxiety. Model 5 shows that age and religion are significantly associated with somatization. Compared to residents in communities with a low COVID-19 severity, residents in communities with a medium and high COVID-19 severity tend to have stronger symptoms of somatization (0.79 and 1.68, respectively). These effects continue to be significant and remain sizable in Model 6, in which other predictors are incorporated. SES has a significant direct impact on somatization; a one-unit increase in SES will decrease somatization by a margin of −0.17. This coefficient becomes insignificant in Model 6, in which the measures of lifestyle and social capital mediate all of SES’s effect. In this model, health-damaging behavior increases one’s somatization by 0.35, health-promoting behavior does not generate a significant coefficient, valuation of individualism decreases one’s somatization by −0.21, network intensity decreases one’s somatization by −0.59, and network extensity decreases one’s somatization by −0.32. The explained variance substantially increases from Model 5 to Model 6, as shown by the corresponding adjusted R-squared values of 0.063 and 0.133. The significant coefficients from models 5–6 provide additional evidence in support of propositions 1–3.

### 4.7. Interactive Effects on Mental Symptoms

Proposition 4 predicts that SES modifies the impact of COVID-19 severity on mental symptoms. The regression results in Table 7 support this prediction.

Model 7 presents the interactive effects on depression. The interaction term between SES and a medium severity of COVID-19 has an insignificant coefficient of −0.28 (*p* = 0.169), but the coefficient for the interaction term between SES and high severity, with a value of −0.96, is highly significant (*p* = 0.001). These coefficients mean that the effect of SES on depression is about the same for communities with low and medium degrees of COVID-19 severity, but it is stronger when the COVID-19 severity increases to a high level, providing partial support for Proposition 4.

Since the results in models 8 and 9 reveal the same patterns as interpreted above, we briefly review these results in turn. Model 8 shows that communities with medium and low degrees of severity of COVID-19 have a similar level for SES’s impact on anxiety (a coefficient of −0.24, *p* = 0.241), but communities with a high degree of COVID-19 severity have a significantly stronger SES impact on anxiety (a coefficient of −0.72, *p* = 0.014). Model 9 exhibits a similar pattern on somatization, with high-severity communities resulting in a significant alteration to SES’s impact when compared to low- and medium-severity communities, by a margin of −0.68 (*p* = 0.001).

Figure 2 sums up the interactive effects through graphic presentation. The straight lines were constructed by using the predicted values of the three dependent variables from the respective regression models, converting the coefficients of interaction terms into values based on the changing values of SES (see Appendix A for technical details). Two summary interpretations surface here. First, across the three mental symptoms, SES has a decreasing impact; the higher one’s SES, the lower one’s symptoms of depression, anxiety, and somatization. Second, SES modifies the impact of COVID-19 when the severity is assessed as high: The regression line for high-severity COVID-19 communities in each panel is much steeper than those for medium- and low-severity COVID-19 communities. This makes it explicit that an increased severity of COVID-19 tends to maximize SES’s effect on mental symptoms. Combined with the findings of Table 7, Figure 2 sets a solid footing for accepting Proposition 4.

## 5. Discussion

The survey of Chinese WeChat networkers in August 2020 has presented us with an impressive set of findings that deserve our attention. First and foremost, the spread of COVID-19 matters in terms of mental health. On a 6–30 scale, an average adult Chinese person scores 9.14 for depression, 8.77 for anxiety, and 7.65 for somatization, based on our study. Compared to the results of an April survey [71], the prevalence of mental symptoms decreased significantly in August; during the four months between the two surveys, the spread of coronavirus was conceivably brought to a minimum level throughout China. Had this comparison been based on a panel study, the results would have indicated the effect size of COVID-19 on mental symptoms. However, since these results were obtained from two separate cross-sectional surveys, and because their studied populations and measurements of mental symptoms were not comparable, the comparison requires further verification. At the same time, the impact of COVID-19 can be ascertained from our own survey. With a low COVID-19 severity as a point of comparison, depression, anxiety, and somatization increase significantly under medium and high levels of COVID-19 severity, with substantial incremental margins (Figure 1 and Table 3). These increment margins are largely maintained in OLS regression models (Table 4, Table 5, Table 6 and Table 7), in which variables of personal attributes are statistically controlled for.

Second, this study focused on three *social mechanisms* of mental symptoms, including SES, lifestyle, and social capital, and our OLS regressions revealed that each of these variables has a significant impact on mental symptoms. This means that the social positions people hold, the lifestyles they pursue, and the resources they mobilize from social relationships all matter for their psychological well-being. While SES’s impact is commonly known among mental health researchers, our regression models treat measures of lifestyle and social capital as the mediators, and they prove that SES’s impact on mental health is indirect and robust through the mediating roles of lifestyle and social capital. This implies that future research will need to simultaneously consider all three of these sociological variables when modeling the roles that social mechanisms play in mental health. While mental health is a human phenomenon, the changing processes of mental health conditions at individual levels are, after all, social.

Third, as a social mechanism of mental symptoms, SES’s impact was contextualized. As shown in our last set of regression models, when a residential community has not only confirmed cases, but also deaths, the severity of the spread of the virus is considered high. Under this situation, SES’s impact is significantly larger. We have interpreted this finding from resource-maximization logic: SES-embedded resources are mobilized to the highest possible extent for reducing potential or visible risks. This logic may be behind the widely reported higher rates of COVID-19 infection among ethnic minorities than the wealthier White majority in the United States and elsewhere [73]. Because COVID-19 severity is not a natural phenomenon, but a result of human mitigation, including governmental interventions and social mechanisms (e.g., lifestyle and social capital), future research should explore institutional, cultural, and political contexts to explain the differential SES impact across levels of epidemic severity.

The above summary results indicate that this study presents several contributions to the study of mental health under COVID-19. First, this study provides a timely assessment of mental health conditions among Chinese people under the mode of normative control and prevention of COVID-19. Second, since this study classified communities as having high, medium, and low levels of COVID-19 severity, our assessment can be used as a benchmark to predict the level of mental health under varying degrees of COVID-19 severity. Third, our statistical models have uncovered the underlying patterns of how SES, lifestyle, and social capital are simultaneously related to mental health under COVID-19, and the observed patterns can be verified, revised, or expanded in future research from both a Chinese and comparative perspective.

We must be cautious about the generalizability of our study results. Our survey was cross-sectional and conducted at a particular time (August 2020), when the threat of COVID-19 was already under good control in China. Therefore, surveys with comparable designs, but conducted at other times, either earlier or later, can be used to evaluate the results of this survey. Additionally, our sample of WeChat networkers is representative of the younger and highly educated segment of China’s adult population. Therefore, the estimated significant and insignificant effects of respondents’ personal attributes, such as gender, age, marital status, and religious belief, as well as our sociological variables, must be re-estimated in future studies by using survey samples more representative of China’s adult population, with enough older and less educated people, who are more vulnerable to epidemic diseases than younger and more educated people. Finally, the time and space constraints of our online survey prevented us from including a full inventory of mental health symptoms, and we were also confined to a small number of explanatory and control variables. Whether these study limitations have affected our reported findings in a conceivable way needs to be verified in future research using a face-to-face mode of survey interviews.

## 6. Conclusions

Mental health conditions in China varied among survey respondents under different levels of COVID-19 severity, and residents in communities with a high severity of the epidemic exhibited more serious symptoms of depression, anxiety, and somatization. It has been revealed that socioeconomic status is directly and indirectly associated with depression, anxiety, and somatization, and the impact of socioeconomic status is maximized under the most severe COVID-19 situation, where both confirmed cases and deaths are reported. Healthy and individualism-oriented lifestyles, as well as bonding and bridging social capitals, mediate the impact of socioeconomic status, and together, these sociological factors jointly reduce the negative impact of COVID-19 on mental health.

## Figures and Tables

**Figure 1 ijerph-17-08843-f001:**
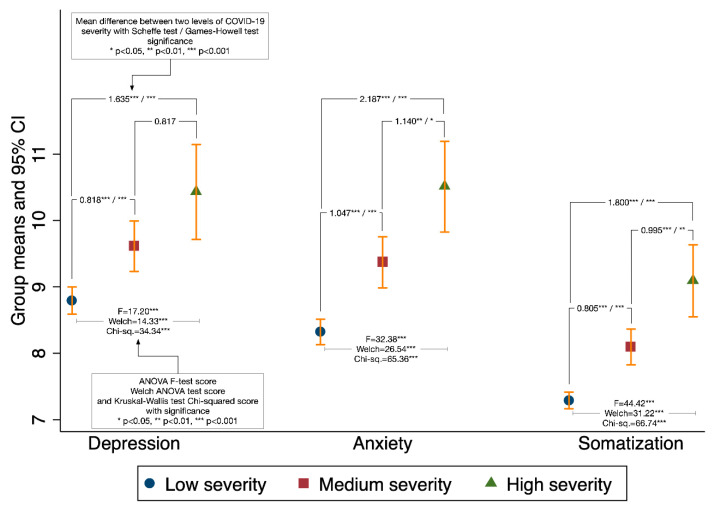
Mean scores of mental symptoms by level of COVID-19 severity.

**Figure 2 ijerph-17-08843-f002:**
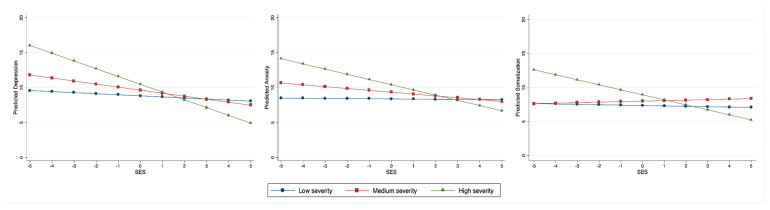
Interactive effects of socioeconomic status (SES) and COVID-19 severity on mental health.

**Table 1 ijerph-17-08843-t001:** Characteristics of survey respondents (*n* = 2015).

Variables	*n* (%) or M ± SD	Variables	*n* (%) or M ± SD
Gender		Education	
Male	993 (49.28)	graduate school	134 (6.65)
Female	1022 (50.72)	college/university	1412 (70.07)
Age (years old)	27.95 ± 7.15	senior high	412 (20.45)
18–29	1263 (62.68)	junior high and below	57 (2.83)
30–39	619 (30.72)	Occupation	
40–49	107 (5.31)	small employer	173 (8.59)
≥50	26 (1.29)	white-collar employee	855 (42.43)
Marital status		blue-collar employee	491 (24.37)
Married	1440 (71.46)	college student	418 (20.74)
Otherwise	575 (28.54)	jobless	78 (3.87)
Religious belief		Family income (yuan)	176,733 ± 232,867
Yes	281 (13.95)	Severity of COVID-19	
No	1734 (86.05)	high severity	175 (8.68)
Residence		medium severity	499 (24.76)
Urban	1112 (55.19)	low severity	1341 (66.56)
Rural	903 (44.81)	Mental symptoms	
CCP membership		depression	9.14 ± 4.08
Yes	358 (17.77)	anxiety	8.77 ± 3.94
No	1657 (82.23)	somatization	7.65 ± 2.73

Note: M = mean; SD = standard deviation; and CCP = Chinese Communist Party.

**Table 2 ijerph-17-08843-t002:** Results of factor analysis: Orthogonal varimax rotation solutions.

Factors and Variables	Factor Loadings	Factors and Variables	Factor Loadings
SES: VE = 53%, KMO = 0.62		Lifestyle: VE = 68%, KMO = 0.53	
Education	0.70	F1: Health-damaging behavior	
Occupation	0.73	Frequency of smoking	0.87
Family income	0.75	Frequency of drinking	0.86
Max = 2.58, min = −3.61		Max = 2.75, min = −1.25	
Social Capital: VE = 61%, KMO = 0.55		F2: Health-promoting behavior	
F1: Network intensity		Physical exercise frequency	0.79
Spousal relations	0.89	Physical checkup frequency	0.77
Intergenerational relations	0.88	Max = 1.72, min = −2.43	
Max = 2.06, min = −3.62		F3: Valuation of individualism	
F2: Network extensity		Sense of privacy	0.80
Online interaction	0.76	Freedom of expression	0.83
Frequency of online activity	0.70	Max = 1.83, min = −2.77	
Number of daily contacts	0.60		
Max = 2.15, min = −9.29			

Note: VE = variance explained, SES = socioeconomic status and KMO = Kaiser–Meyer–Olkin measure of sampling adequacy.

**Table 3 ijerph-17-08843-t003:** Standardized effect sizes of mental health across levels of COVID-19 severity.

	Group Mean			Group S.D.			Cohen’s d		
Measures	${\bar{x}}_{low}^{k}$	${\bar{x}}_{medium}^{k}$	${\bar{x}}_{high}^{k}$	$S_{low}^{k}$	$S_{medium}^{k}$	$S_{high}^{k}$	$d_{high-low}^{k}$	$d_{medium-low}^{k}$	$d_{high-medium}^{k}$
Depression	8.79	9.61	10.43	3.82	4.31	4.83	0.42	0.21	0.18
Anxiety	8.32	9.37	10.51	3.58	4.37	4.61	0.59	0.28	0.26
Somatization	7.29	8.10	9.09	2.34	3.07	3.66	0.71	0.32	0.31

Note: $n_{low}=1341$; $n_{medium}=499$; $n_{high}=175$; S.D. = standard deviation.

**Table 4 ijerph-17-08843-t004:** Ordinary Least Square (OLS) regression results on depression.

Variables	Model 1		Model 2	
	Β (95% CI)	*p*-Value	Β (95% CI)	*p*-Value
Gender	0.17 [−0.18, 0.52]	0.334	0.09 [−0.26, 0.45]	0.608
Age	−0.04 [−0.07, −0.01]	0.004	−0.05 [−0.07, −0.02]	0.001
Marital status	−0.95 [−1.38, −0.53]	0.000	0.20 [−0.25, 0.65]	0.387
Religious belief	0.32 [−0.19, 0.83]	0.224	0.35 [−0.13, 0.84]	0.150
Residence	0.09 [−0.29, 0.46]	0.652	0.13 [−0.22, 0.49]	0.464
CCP membership	−0.20 [−0.67, 0.27]	0.403	0.06 [−0.39,0.51]	0.793
Medium severity (vs. low)	0.97 [0.57, 1.38]	0.000	0.82 [0.43, 1.21]	0.000
High severity (vs. low)	1.81 [1.18, 2.44]	0.000	1.47 [0.87, 2.08]	0.000
SES	−0.52 [−0.72, −0.33]	0.000	−0.30 [−0.49, −0.11]	0.002
Health-damaging behaviors			0.31 [0.12, 0.49]	0.001
Health-promoting behaviors			−0.40 [−0.57, −0.22]	0.000
Values of individualism			−0.22 [−0.39, −0.06]	0.007
Network intensity			−1.24 [−1.43, −1.05]	0.000
Network extensity			−0.34 [−0.51, −0.18]	0.000
Constant	10.41 [9.64, 11.18]	0.000	9.79 [9.03, 10.56]	0.000
*n*	2015		2015	
Adjusted *R*^2^	0.063		0.167	

Note: Confidence intervals are shown in square brackets. Dummy variables: Gender (male = 1, female = 0); residence (urban = 1, rural = 0); and the remaining dummy variables (yes = 1, no = 0).

**Table 5 ijerph-17-08843-t005:** OLS regression results on anxiety.

Variables	Model 3		Model 4	
	Β (95% CI)	*p*-Value	Β (95% CI)	*p*-Value
Gender	−0.07 [−0.41, 0.27]	0.688	−0.19 [−0.54, 0.17]	0.296
Age	−0.05 [−0.07, −0.02]	0.001	−0.05 [−0.08, −0.03]	0.000
Marital status	−0.22 [−0.63, 0.19]	0.296	0.62 [0.17, 1.06]	0.007
Religious belief	0.62 [0.12, 1.11]	0.015	0.61 [0.13, 1.09]	0.013
Residence	0.03 [−0.34, 0.39]	0.883	0.05 [−0.31, 0.41]	0.790
CCP membership	−0.26 [−0.72, 0.20]	0.261	−0.09 [−0.53, 0.36]	0.705
Medium severity (vs. low)	1.10 [0.71, 1.50]	0.000	0.96 [0.57, 1.35]	0.000
High severity (vs. low)	2.18 [1.57, 2.80]	0.000	1.90 [1.30, 2.50]	0.000
SES	−0.28 [−0.47, −0.09]	0.004	−0.14 [−0.33, 0.05]	0.137
Health-damaging behaviors			0.29 [0.11, 0.48]	0.002
Health-promoting behaviors			−0.15 [−0.32, 0.03]	0.098
Values of individualism			−0.28 [−0.44, −0.12]	0.001
Network intensity			−0.97 [−1.16, −0.78]	0.000
Network extensity			−0.27 [−0.44, −0.11]	0.001
Constant	9.74 [8.99, 10.50]	0.000	9.38 [8.62, 10.15]	0.000
*n*	2015		2015	
Adjusted *R*^2^	0.051		0.115	

Note: Confidence intervals are shown in square brackets. Dummy variables: Gender (male = 1, female = 0); residence (urban = 1, rural = 0); and the rest of the dummy variables (yes = 1, no = 0).

**Table 6 ijerph-17-08843-t006:** OLS regression results on somatization.

Variables	Model 5		Model 6	
	Β (95% CI)	*p*-Value	Β (95% CI)	*p*-Value
Gender	0.07 [−0.16, 0.31]	0.533	−0.13 [−0.37, 0.11]	0.290
Age	−0.02 [−0.04, −0.00]	0.023	−0.03 [−0.05, −0.01]	0.002
Marital status	0.06 [−0.23, 0.34]	0.702	0.48 [0.17, 0.78]	0.002
Religious belief	0.94 [0.60, 1.28]	0.000	0.90 [0.57, 1.23]	0.000
Residence	−0.06 [−0.31, 0.20]	0.662	−0.04 [−0.28, 0.21]	0.766
CCP membership	−0.02 [−0.34, 0.29]	0.879	0.06 [−0.24, 0.37]	0.693
Medium severity (vs. low)	0.79 [0.52, 1.07]	0.000	0.67 [0.41, 0.94]	0.000
High severity (vs. low)	1.68 [1.26, 2.10]	0.000	1.42 [1.01, 1.83]	0.000
SES	−0.17 [−0.30, −0.04]	0.009	−0.09 [−0.22, 0.04]	0.166
Health-damaging behaviors			0.35 [0.22, 0.48]	0.000
Health-promoting behaviors			−0.05 [−0.17, 0.07]	0.376
Values of individualism			−0.21 [−0.32, −0.09]	0.000
Network intensity			−0.59 [−0.72, −0.46]	0.000
Network extensity			−0.32 [−0.43, −0.21]	0.000
Constant	7.74 [7.22, 8.26]	0.000	7.75 [7.22, 8.27]	0.000
*n*	2015		2015	
Adjusted *R*^2^	0.063		0.133	

Note: Confidence intervals are shown in square brackets. Dummy variables: Gender (male = 1, female = 0); residence (urban = 1, rural = 0); and the rest of the dummy variables (yes = 1, no = 0).

**Table 7 ijerph-17-08843-t007:** Interaction effects from OLS regressions on mental symptoms.

Variables	Model 7 (Depression)		Model 8 (Anxiety)		Model 9 (Somatization)	
	Β (95% CI)	*p*-Value	Β (95% CI)	*p*-Value	Β (95% CI)	*p*-Value
Gender	0.09 [−0.26, 0.45]	0.608	−0.19 [−0.55, 0.16]	0.293	−0.12 [−0.36, 0.12]	0.332
Age	−0.05 [−0.07, −0.02]	0.000	−0.05 [−0.08, −0.03]	0.000	−0.03 [−0.05, −0.01]	0.002
Marital status	0.21 [−0.23, 0.66]	0.349	0.63 [0.18, 1.08]	0.006	0.47 [0.17, 0.78]	0.003
Religious belief	0.39 [−0.09, 0.87]	0.114	0.64 [0.16, 1.12]	0.009	0.91 [0.59, 1.24]	0.000
Residence	0.14 [−0.21, 0.50]	0.430	0.06 [−0.30, 0.41]	0.756	−0.04 [−0.28, 0.21]	0.777
CCP membership	0.09 [−0.36, 0.53]	0.701	−0.07 [−0.51, 0.38]	0.775	0.08 [−0.23, 0.38]	0.622
Medium severity (vs. low)	0.82 [0.43, 1.21]	0.000	0.96 [0.57, 1.35]	0.000	0.65 [0.38, 0.91]	0.000
High severity (vs. low)	1.66 [1.04, 2.27]	0.000	2.03 [1.42, 2.64]	0.000	1.58 [1.16, 2.00]	0.000
SES	−0.15 [−0.37, 0.06]	0.168	−0.02 [−0.24, 0.19]	0.828	−0.06 [−0.20, 0.09]	0.453
Health-damaging behaviors	0.31 [0.13, 0.50]	0.001	0.30 [0.11, 0.48]	0.002	0.35 [0.22, 0.47]	0.000
Health-promoting behaviors	−0.41 [−0.58, −0.23]	0.000	−0.15 [−0.33, 0.02]	0.083	−0.06 [−0.18, 0.06]	0.330
Values of individualism	−0.23 [−0.39, −0.06]	0.007	−0.28 [−0.44, −0.12]	0.001	−0.21 [−0.32, −0.09]	0.000
Network intensity	−1.25 [−1.44, −1.06]	0.000	−0.97 [−1.16, −0.78]	0.000	−0.59 [−0.72, −0.46]	0.000
Network extensity	−0.34 [−0.51, −0.18]	0.000	−0.27 [−0.44, −0.11]	0.001	−0.32 [−0.43, −0.20]	0.000
SES × Medium severity	−0.28 [−0.68, 0.12]	0.169	−0.24 [−0.64, 0.16]	0.241	0.13 [−0.14, 0.40]	0.352
SES × High severity	−0.96 [−1.54, −0.38]	0.001	−0.72 [−1.30, −0.14]	0.014	−0.68 [−1.08, −0.29]	0.001
Constant	9.82 [9.06, 10.59]	0.000	9.41 [8.64, 10.17]	0.000	7.75 [7.23, 8.27]	0.000
*n*	2015		2015		2015	
Adjusted *R*^2^	0.171		0.117		0.138	

Note: Confidence intervals are presented in square brackets. Dummy variables: Gender (male = 1, female = 0); residence (urban = 1, rural = 0); and the rest of the dummies (yes = 1, no = 0).

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
