# Peer review of "Mental Health of Chinese Online Networkers under COVID-19: A Sociological Analysis of Survey Data"

_ijerph, 2020, doi:10.3390/ijerph17238843_

Round 1

Reviewer 1 Report

Dear authors,

This paper needs a lot of work in my opinion, but it could be of interest to readers if the authors will do this work.

Please find comments on your paper below.

"The cited study is silent on mental health, however. Mental health research from around the world, on the other hand, has revealed that socioeconomic status is a consistent source of variation in a stream of mental health symptoms [4,5], such as depression, anxiety, and somatization [6,7]."

This seems like a crypto-causal claim. If you want to make a causal claim of social status, then please provide some causal evidence. If you want to just claim that social status is statistically associated with mental health, make this more clear. See e.g., this paper on the use of causally ambiguous language in social science. In my opinion, it is best avoided.

https://psyarxiv.com/nkf96/

“Does SES primarily affect health (the social causation hypothesis) or does health primarily affect SES (the health selection hypothesis)?”

They can also have common causes (both caused by Z). On causal evidence on socialeconomic status and health, see these interesting studies of intra-individual data which suggests income is not causal for health, as least insofar as self-reported health goes. This paper provides a recent, overly optimistic review (in my opinion) https://www.sciencedirect.com/science/article/pii/S2352250X19300521, check out some of the primary studies: https://pubmed.ncbi.nlm.nih.gov/25260937/ https://jech.bmj.com/content/69/9/899

Because of evidence like the above, I don’t think it is wise to assume causal effect of SES on mental health (as in your Figure 1 path model). I would guess it runs mostly in the other direction (mentally ill people cause many problems for themselves in life, are bad at keeping jobs, or completing educations etc., all of which causes lower SES index scores).

Additionally, I think that it probably makes more sense to indicate the interaction effect (4) as SES modifying the effect of COVID-19 impact. If one is wealthy, it is a lot easier to handle a sudden job loss from COVID-19, which might otherwise lead to severe stress and depression.

“With the professional assistance of a WeChat survey company based in Shenzhen neighboring

Hong Kong, we designed a short module for an average survey time of ten minutes.”

Please provide more details of this. Why did the subjects take part in this survey? Were they paid or had a chance of some award, or just plain volunteers? How representative is this sample compared to your target population? You later say it isn’t entirely representative. You should present some numbers to see how much. E.g., what is the average age of the population you want to generalize to? If you are concerned about an age interaction, you could add a three-way interaction model. In general, one might expect age is related to anxiety, as older people are much higher risk of death from COVID. So, there should be an interaction for age*severity.

“detective questions”

More common term is attention checks.

“As shown in Table 1, in terms of both mean and standard deviation, depression is a stronger symptom with a wider dispersion (9.14 + 4.08) than anxiety (8.77 + 3.94) and somatization (7.65 + 2.73). This indicates that depression is more a serious mental health problem than anxiety or somatization in China during the Covid-19 pandemic.”

This inference seems to be based on the implicit assumption that these scales follow the same scale just because they have the same number of items (6). It could be that 5 on one scale corresponds to average non-COVID levels while 7 on the other scale corresponds to the non-COVID average. You need a comparison group here.

“We transformed the factorial score into a 1-100 scale for the purpose of easy interpretation. The higher the score, the higher the respondent’s SES.”

I don’t think this is easier to interpret than z scores. When you do this min-max standardization, you rely upon the sample’s min and max scores, which are unreliable metrics. I think better here to just use the z scores, which will be roughly comparable to other studies. You can also compute centile scores, if you want a 0-100 scale.

I am not sure it is wise to transform the variables to 1-100 scale, and then compare the standard deviations (SD). It may be better to do this on the original raw scale. I am not totally sure though. Maybe see if you can find advice somewhere on this.

In any case, your differences in SDs are quite small, and possibly not reliable. You can use the ‘variance ratio test’ for this purpose. https://link.springer.com/chapter/10.1007/978-981-13-0827-7_12 However, it may be wiser to do this comparison using bootstrapping so you don’t have to make parametric assumptions.

“A factor analysis confirms this 2-dimension solution”

Please provide details of this. It sounds like you used some default settings. If you use an oblique rotation, please note the factor correlations as well.

Again, your 1-00 transformations are probably unwise. What you are doing here is showing that the scale of scores is wider on some scales. This can be due to outliers, not necessarily more variation as such.

“age-squared (to detect a possible curve-linear age effect on mental health)”

Generally speaking, it is better to use splines for this (natural splines best, in my opinion). You don’t seem to have any particular reason to assume a 2nd degree polynomial form. The spline approach doesn’t make this assumption.

“(p-value=0.000)”

P values cannot be 0. You mean “<0.001” probably. This error occurs many times, e.g. Table 2.

“refers to a community in which no confirmed Covid-19 cases were identified”

Where are these data from? Which units?

I suggest you plot the distributions of the mental health scales by severity. This will show the reader better the effect size. You should also calculate the standardized effect sizes (especially Cohen’s d between groups). These are needed for meta-analysis.

“Proportional change with low severity as base”

You cannot do ratio operations on non-ratio scale data. Your data are interval scale at best. https://en.wikipedia.org/wiki/Level_of_measurement If you want to do something like this, you can designate an arbitrary value as a threshold to recode the data into binary form (0/1), which is a ratio scale, and would allow you to calculate the ratio increase in the prevalence of the mental illness, insofar as defined by that threshold (different thresholds will produce different results).

Note that this error occurs later as well e.g. “residents in communities with a medium Covid-19

severity have higher depression by a margin of 0.90 (p-value=0.000), or 8.6% (0.90/10.45)” (one cannot compute percentages, these involve division, and one cannot use division without a true zero).

“To save space, the coefficients of these control variables are omitted from the table (full results are available upon request)”

This is not good enough. Please put the full results in the supplement. I also strongly suggest you upload your code/syntax, statistical output, and data to a science repository. You can use https://osf.io/ which is free and easy to use. Remember to remove personally identifying information from the dataset before uploading it.

“Nearly all control variables generate no significant coefficients, indicating that Chinese people tend to have a similar level of depression even if they vary by gender, age, religion, residence, and party affiliation (except for marital status, a significant predictor).”

This is a red flag to me. All these variables are known to relate to mental illness. It is very strange that you find no relationships. You may want to re-examine the data and be sure of this conclusion! For instance, sex differences in depression are extremely well-documented. https://www.ncbi.nlm.nih.gov/pmc/articles/PMC5532074/

“Model 2 includes SES as an additional predictor of depression. It shows that for a one-unit

increase in SES (a 1-100 scale), depression will decrease by a margin of -0.04 (p-value=0.000). Although the effect size seems small, it is substantial. Consider the following scenario: As compared to an average Chinese person with a mean SES of 58.79 and a standard deviation of 16.0 (Table 1) and a mean depression of 9.64 (the constant in this model), another person of equal attributes but having one standard deviation above the mean SES will decrease his/her depression by a margin of 0.64 (0.04 x 16), or a 6.6% (0.64/9.64) drop on the depression scale from its mean; and still another person of equal attributes but having two standard deviations below the SES mean will increase his/her depression by a margin of 1.28 (0.04 x 32), or a 13.2% jump on the depression scale from its mean. Since depression is a stable scale across a person’s life course [20], the coefficient of -.04 reported here must be taken seriously. This brings support to Proposition 2.”

While one can keep variables on these scales, if you converted the numerical variables to z-scores, these coefficients would be a lot easier to interpret (standardized betas).

“Table 6. Interaction Effects from OLS Regressions on Depression, Anxiety, and Somatization”

I note that the addition of your interaction effects does not affect the model R2 very much. Eg., compare Model 9 in Table 5, R2=13.9%. Model 12 (same with interaction) R2 = 14.5%. Are these R2 values adjusted for overfitting (“adjusted R2) or are they regular R2? You should use the adjusted ones.

Whatever the ase, you should discuss this finding. Most of the model validity is from the main effects, not the interactions. I like the plots in Figure 2. They help interpret the model results.

Your model results are very strange. There is no main effect of SES, but you find an interaction with a tiny p value. This is a yellow flag. Interaction effects without main effects are suspicious. If you search around a bit, you will find a lot of discussion of this.

“three social suppressors of mental health symptoms:”

Beware that a suppressor has another meaning in statistics. https://stats.stackexchange.com/questions/73869/suppression-effect-in-regression-definition-and-visual-explanation-depiction

“It has been revealed that socioeconomic status is an effective social suppressor of depression, anxiety, and somatization, and this suppressor’s impact is maximized under the severest Covide-19 situation.”

This conclusion is too far. Your results are in line with this causal model, but it could be other stuff. Maybe SES is a proxy for other factors, e.g. general health or intelligence or personality aspects, that are the real causes.

Reviewer 2 Report

Thank you for inviting me to review this manuscript regarding the impact of Covid19 on the mental health of the general population. Obviously, studies like this are very timely and highly needed. It is also interesting that the study adopted a sociological perspective, which may allow interpretation of the data on a broader, structural level. Despite the importance of the topic, the paper includes several unclarities. However, there are two major, and crucial shortcomings that must be addressed before the paper can be considered for publication.

First, there is unclarity on how the severity or the impact of Covid19 was conceptualized and measured. It was not addressed in the introduction, and the description in the methods section is absolutely vague.

Second, Throughout the paper, the authors refer to a causal relationship between study findings of Covid19 and other outcomes. Just one example: ‘These results mean that the Covid-19 pandemic indeed had brought psychological problems to Chinese citizens’. As this is a cross-sectional study, all statements about causality should be rephrased throughout the manuscript.

Also, statements in the conclusion (‘Mental health conditions in China were significantly deteriorated by the Covid-19 pandemic’) cannot be made based on this study.

Further comments are listed below.

Abstract

The abstract could mention the Method of the study.

Please have the manuscript proofread by a native English speaker. There are several awkward wordings and formulation. For example:

Page 1, line 27: ‘ordinary people’, could be members of the general public, or something similar.

Line 100-101: “To be sure, people with higher SES tend to have greater social capital stocks”: is not clear.

Line 107: what is ‘network generated’ social capital?

Line 110: The text suggests that social capital can affect mental health, irrespective of, or above, SES status. Is that correct? Could you be more specific or clarify?

Page 4, proposition 1: this is not clear.

First the text says that mental health is affected by: ‘the severity of the Covid-19 pandemic’. Next it says that it is affected ‘more severely by the outbreak of Covid-19’. These seem to be two different ‘things’, and the text has not provided definitions of either of them.

It is also not in line with the ‘severity of Covid19’ in the method section.

A similar comment applies to proposition 4: what is ‘Covid-19 severity’?

In ‘Materials and methods’ it was specified that ‘WeChat’ users are the ‘socioeconomically most active population’. Obviously, this could constitute a serious bias to the study, especially since the study wants to examine the effect of SES.

Line 161: What do you mean with ‘left free’?

Line 163: The text stated that Tibet is a province of China. I am not sure if all international readers will agree with this political statement.

And how come there were no participants from this ‘province’?

Line 180: You cannot say that it was more important ‘during the Covid-19 pandemic’. You can only say that it was more important at the time of the data collection. This was not a longitudinal study.

Lines 182-190: The text is not very clear. However, I understand that these were non-standardised cut-off scores. This would entail a serious shortcoming to the study, which should be reported in the limitations.

Page 6, lines 193-199: This paragraph is absolutely not clear, and constitutes a crucial weakness of the study.

What is a ‘residential community’?

What is the source of the data? Self-report from participants? Official records? Reliability?

What was the timeframe of ‘no cases’ or ‘no deaths’?

It is also not clear how adequate this measure is. Can you measure the severity of Covid19 based on the number of cases? What about the speed of the spread of the virus? Or people having family or friends getting affected? What about exposure to ‘news’ about the virus, either through the official media channels or informal channels?

Overall, this variable requires a much more solid definition and operationalization.

Page 7, Analyses

Please explain that the statistical requirements to conduct ANOVA and regressions have been met.

What is OLS? Please write OLS in full the first time that you use this abbreviation.

Given the crucial comments mentioned above, I will not go into the detail of the findings and discussion. I wish the authors good luck with revising the manuscript.

Reviewer 3 Report

Dear authors,

This is, overall, a good manuscript, that summaries the findings of a considerably extensive research carried out in China. Firstly, and bearing in mind the current state-of-affairs of COVID19 over general and working populations, the topic seems to be interesting and worth of investigation.

However, there are some major issues that need further clarification, consideration and revisions from the authors. Please see below:

  • Abstract:

The abstract of the paper requires further structuration, in terms of both form and contents. The current section is very modest and does not properly summarize the contents and the richness of the study. For instance, the aim or purpose of the study remains unclear, the sample could be better described (in sociodemographic terms) and a bit of background (2-3 lines) about the Chinese panorama at COVID outbreak and its health/welfare implications might help the readers to develop a more holistic impression on the paper.

  • Introduction:

The introduction has been adequately structured, providing useful information on the possible relationships among population features and different mental health-related consequences of the pandemic.

Further, the theoretical model seems to be well-structured and founded on the available evidences (that, although scarce, are interesting), proposing some plausible paths to explain mental health outcomes on the population. However, the operational definition of “Covid-19” needs more explicitness, in accordance to the measures used in the study, that are just visible at the methods.

Also, and before raising up the study hypotheses, authors must present the study aim in a more adequate way, in order to give more pertinence to the hypotheses proposed by the authors.

  • Methods:

Although the sample is considerably large, you are not referring to the overall Chinese population; in fact, the sample cannot be representative of the populational frame, since it refers, rather, to a very specific population segment. Accordingly, authors should a) state this consideration at the section 3.1, rescope the title of the study (for instance, by adding a scoping resource, such as “(…)among Chinese Networkers”), and in-depth discuss (perhaps at the limitations) its implication on the validity and generalizability of the study.

Also, and as aforementioned, the variable “COVID-19” (severity; scored by respondents) seems to be not properly operationalized, and its definition is very blurred. In this regard, authors must remark if it was whether a “perceived severity” measure, or a fixed parameter established by them (and how).

In regard to variables such as “social capital” and “lifestyle” it could be interesting to be able to review the criteria (i.e. items and scales) used to assess these constructs among participants. Authors describe the subsequent FAs (Factor Analyses), but not knowing the items and their meaning makes difficult to validate the data.

  • Results:

In regard to Table 1, it is worth highlighting. That authors used the BSI-18, that is based on a Likert scale. Thus, the impression is that these participants were virtually diagnosed, while the scope of the questionnaire is only to perform a screening on possible symptoms present in these three factors (so, we don’t talk about “levels of depression” but about its symptomatology, as actually allowed by the short questionnaire. Although it does not invalidate the study or the measure itself, authors should be more careful to report this data.

The rest of the results seem to be well-organized and to have followed a good analytic strategi to assess the intended relationships.

  • Discussion:

From my perspective, discussion is good. Authors carefully describe the main findings, referring to similar highlights provided by the scientific literature appended in the introduction. However, there are two issues that need the attention of authors:

  1. Authors could hypothesize a bit more about the changes described at lines 399-401. Also, the paragraph between lines 407-414 could be better discussed by the authors, since the ideas contained are quite interesting, but need more support.
  2. At the limitations, authors could address the temporal shortness of the study; in other words, this data collection took place at a certain moment of the pandemic, where the context was quite particular and differential if compared to previous or ulterior stages of it. As I understand it exceeds the ability of any researcher, it should be, at least, appended as a limitation that might influence the results of the study.

  • Conclusion:

Unlike discussion, I found the first conclusion of the paper a lot outside the actual scope of the study. This fact is not only tautological, but it has not been studied at the present research. Hence, authors should refer to the study findings, in accordance to the study aim, that (as aforementioned) remained very unspecific at the first version of the paper.

Reviewer 4 Report

The manuscript "Mental Health under COVID-19 in China: A Sociological Analysis of Survey Data” describes a study evaluating the impact of socioeconomic status during the COVID-19 pandemic on mental health in a WeChat networkers’ sample.

The question posed by the authors is well defined in the introduction and appears to fit the aims and scope of the journal. The design of the study is clear and the methods are well described. The statistical analyses are appropriate to the study design. The results are adequately presented, the discussion is exhaustive and the conclusions correlate to the findings.

I think the article is suitable for publication already in the present form.  

However, I would just suggest to the authors to describe Table 1 in the Results section. The first paragraph of the Results section could fully describe the characteristics of survey respondents (e.g. lines 177 to 181, 198-199, 218 to 220…).  

Reviewer 5 Report

This study from China shows very clear results in the analysis of complex interactive relationships between  SES, community exposure to covid-19 and mental health outcomes depression, anxiety and somatization. Their basic conclusion seems reasonable - that covid-19 hits harder community-wise with regard to mental health when social capital and socioeconomic group are low.

The study sample was recruited from WeChat networkers. It is not clear how the participants were invited to this particular study and how the authors related to ethical questions. They state that

… Two detective questions were used in the middle of questionnaire to screen out computer-generated, program- coded answers intended to collect a payment.

This means that they tried to avoid non-serious participants. I am not familiar with this technique so it is hard for me to judge whether they may have succeeded. The authors make the comment that these networkers represent China´s socioeconomically most active population. We have to remember this when interpreting the results – they may not be generalizable to the total population. Perhaps this particular group of participants deviate from the general population. Using more conventional criteria we do not know the participation rate either. 

The general background is described in an interesting way. I notice that despite the fact that the participants are socioeconomically "most active" there is nothing at all about their working conditions. Another difference from typical anglosachsan or European publications in this field is that the authors use only two types of possible intermediary variables, health damaging (smoking and drinking)/health promoting (physical exercise and frequency of regular physical check-ups) life style and individualism (valuation of privacy and freedom of expression). A third difference from « our » research tradition is that the authors do not seem to take into account that there are also direct pathways (psychoendocrine) from adverse life conditions to ill health. See for instance Marmot´writings on possible psychoendocrine pathways between low socioeconomic group and health (not only mental by the way)

Among the control variables I note membership in the party. For a European like me this is a highly unusual kind of variable but I do understand that it may indeed be important. This should be discussed more extensively.

There are also differences in ways of presenting data. In figure 2 we are looking at interaction analyses. There are dots placed on perfect straight regression lines. My assumption is that these are illustrating the equations that have been computed, not the raw data so to speak. This is confusing because it is hard to believe that the “real” data look like that. In a sample of 2000 participants it is just not possible that the raw data would look like that. It has to be explained in a better way.

Round 2

Reviewer 1 Report

Dear authors,

I am pleased with the manuscript revisions. You have my approval.

Twi last things. I note that your KMO values are kinda small. Searching around, seems that statisticians suggest values of .50 or .60 as useful thresholds. Your values are close to these, so maybe you are OK. Maybe check out this potential problem.

Your paper is not clear on which factor analysis method was used to obtain the solutions. E.g., a 2 factor solution, you must have used some method that allows for deciding how many factors to retain, and whether these can be correlated or not (oblique vs. orthogonal), and rotated or not (e.g., oblimin). You should clarify this.

Reviewer 2 Report

Dear authors,

Thank you for addressing all my comments. I have no further questions.

Best wishes,

Reviewer 3 Report

Dear Authors,

Firstly, I would like to highlight the adequacy of the amendments and rationales offered in regard to my previous comments. From my opinion, your responses have a good level of detail, and cover well what was expected from you and from the study itself.

This second version of the paper is (as expected) much better than the previously submitted one, considering some particular issues already comprised in the current form of the manuscript. Particularly:

  • The abstract has been edited and presents more and better information on the Chinese context during this early phase of the COVID-19 outbreak.
  • The target population and some of their main features are, actually, better developed and presented now.
  • Authors have corrected a lot of the method-related and intepretative issues impacting the scientific soundness of the paper.
  • Hypotheses are now clear and presented coherently to the methods used.
  • Results are fairly described, also related to the improvement of the variable definitions, measurement considerations and operationalizations.
  • Discussion and limitations were substantially improved, even though the first were already very accurate.
  • Specifically regarding study limitations, the authors did well by warning the readers about the limited (geographical and temporal) capability of generalizing the results of the study. Although it may sound obvious in the first light, this was a necessary step to perform before considering the results as “valid” it selves as a consequence of the fact that the sample was considerably large in this research.
  • Conclusions are more scoped and framed into the actual data derived from the study.

Hence, I would like to say that I have no further comments. Thanks for your responses and all the efforts put on the paper revision.

Best regards.

Reviewer 5 Report

You have handle the comments well